# Metabolome and Mycobiome of *Aegilops tauschii* Subspecies Differing in Susceptibility to Brown Rust and Powdery Mildew Are Diverse

**DOI:** 10.3390/plants13172343

**Published:** 2024-08-23

**Authors:** Veronika N. Pishchik, Elena P. Chizhevskaya, Arina A. Kichko, Tatiana S. Aksenova, Evgeny E. Andronov, Vladimir K. Chebotar, Polina S. Filippova, Tatiana V. Shelenga, Maria H. Belousova, Nadezhda N. Chikida

**Affiliations:** 1All-Russia Research Institute for Agricultural Microbiology, Podbelskogo hwy 3, Pushkin, 196608 St. Petersburg, Russia; chizhewskaya@arriam.ru (E.P.C.); 2014arki@gmail.com (A.A.K.); t.aksenova@arriam.ru (T.S.A.); e.andronov@arriam.ru (E.E.A.); 2St. Petersburg North-West Centre of Interdisciplinary Researches of Problems of Food Maintenance, Podbelskogo hwy, 7, Pushkin, 196608 St. Petersburg, Russia; tipolis@yandex.ru; 3Federal Center N. I. Vavilov All-Russian Institute of Plant Genetic Resources, Bolshaya Morskaya Street, 44, 190121 St. Petersburg, Russia; tatianashelenga@yandex.ru (T.V.S.); m.h.belousova@mail.ru (M.H.B.); n.chikida@mail.ru (N.N.C.)

**Keywords:** *Aegilops tauschii*, seed mycobiome, seed metabolome, leaf rust, powdery mildew, susceptibility

## Abstract

The present study demonstrated the differences in the seed metabolome and mycobiome of two *Aegilops tauschii* Coss accessions with different resistance to brown rust and powdery mildew. We hypothesized that the seeds of resistant accession k-1958 *Ae. tauschii* ssp. *strangulata* can contain a larger number of metabolites with antifungal activity compared with the seeds of susceptible *Ae. tauschii* ssp *meyeri* k-340, which will determine differences in the seed fungal community. Our study emphasizes the differences in the seed metabolome of the studied *Ae. tauschii* accessions. The resistant accession k-1958 had a higher content of glucose and organic acids, including pyruvic, salicylic and azelaic acid, as well as pipecolic acids, galactinol, glycerol and sitosterol. The seeds of *Ae. tauschii*-resistant accession k-1958 were found to contain more active substances with antifungal activity. The genera *Cladosporium* and *Alternaria* were dominant in the seed mycobiome of the resistant accession. The genera *Alternaria*, *Blumeria* and *Cladosporium* dominated in seed mycobiome of susceptible accession k-340. In the seed mycobiome of the resistant k-1958, a higher occurrence of saprotrophic micromycetes was found, and many of the micromycetes were biocontrol agents. It was concluded that differences in the seed metabolome of *Ae. tauschii* contributed to the determination of the differences in mycobiomes.

## 1. Introduction

Microorganisms that inhabit different ecological niches of plants form a plant-associated microbiome that maintains plant stability as they interact with the environment [1]. Plant genes control interactions with associated microorganisms [2,3]. The formation of the plant microbiome, including the fungal community (mycobiome), is influenced by secondary metabolites produced by plants [4,5,6,7]. Plant exudates regulate microbiome assembly through the direct effects of stimulating or inhibiting specific groups of microorganisms [8]. Microorganisms, in turn, can alter the synthesis of plant secondary metabolites, including active metabolites that affect the microbiome [9,10,11].

The microbiome of seeds allows plants to gain advantages in growth and development and defense against biological stresses [12]. The diversity of the fungal community helps plants to adapt more flexibly to changing environmental conditions [13,14,15]. Microorganisms in the microbiome are assumed to be structured and show some phylogenetic organization [16]. However, what constitutes a “healthy” and “beneficial” seed microbiome [12] and how the seed microbiome influences the transmission of fungal diseases to plants in the next generation have not yet been elucidated. In addition, the seed microflora is relatively poorly understood compared to the rhizosphere microflora [17].

The species of the genus *Aegilops* L. are close in evolutionary and taxonomic relations to species of the genus *Triticum* L. *Ae. tauschii* Coss grows in the Mediterranean, in Crimea, in Caucasus, in Central and Small Asia, in Iran and at the foot of the Himalayas. Traditionally, *Ae. tauschii* is divided into two subspecies: *Ae. tauschii* ssp. *tauschii* and *Ae. tauschii* ssp. *strangulata* [18]. The *Ae. tauschii* population is heterogeneous [19]. Using an amplified fragment length polymorphism (AFLP) analysis based on the genetic variability of nuclear genomes, it was shown that the *Ae. tauschii.* ssp. *strangulata* of different origins was divided into two large lineages, with the division not coinciding with the classification division into subspecies [19]. *Ae. tauschii* ssp. *strangulata* is a donor of the D genome of soft wheat *T. aestivum* L. and a carrier of a number of valuable breeding traits, including wheat resistance genes to leaf rust and powdery mildew [20,21,22]. The powdery mildew resistance gene Pm58 (from *Ae. tauschii*) was introgressed into hexaploid wheat and confirmed to be effective under field conditions [20].

Recently, a resistance gene (from *Ae. tauschii*) to leaf rust (*Puccinia triticina* Erikss.) Lr 42 was cloned, showing efficacy at all stages of plant growth [22]. The cloned resistance genes can be used in the assembly of transgenic multigene cassettes to develop resistant cultivars for controlling fungal pathogens [23].

Brown rust (causal agent is the micromycete *Puccinia recondita* Rob. ex Desm f. sp. *triticina* Eriks.) is one of the common and damaging diseases affecting wheat and other cereal crops [24,25,26], as well as powdery mildew (*Blumeria graminis*) [26,27]. These fungal diseases may reduce plant yield by 10–60%, depending on climatic conditions [26,27]. 

It is known that, among the accessions of *Ae. tauschii* species, there are accessions resistant and susceptible to *Puccinia recondita* [28]. 

The accessions of *Ae. tauschii* are stored in the collection of N.I. Vavilov All-Russian Research Institute of Plant Industry (VIR) and every 4–5 years are introduced at the experimental station of VIR to renew the reproduction of the collection accessions and to study plant properties. We selected two accessions from the VIR collection that differ in regard to their susceptibility to leaf rust and powdery mildew.

We hypothesized that the metabolome of k-1958 *Ae. tauschii* ssp. *strangulata*, resistant to brown rust and powdery mildew, would have higher concentrations of metabolites with fungicidal activity and those involved in systemic resistance than the pathogen-susceptible k-340 *Ae. tauschii* ssp. *meyeri*. The mycobiome of the resistant accsession will probably be represented by fewer pathogenic micromycetes and a greater presence of antagonist fungi.

A comparative analysis of metabolomic and microbiome profiles of *Ae. tauschii* seed accessions with different resistance to brown rust was performed in this work.

## 2. Results

### 2.1. Metabolite Profiles of Ae. tauschii Seeds

The seed metabolomes of resistant (k-1958) and susceptible (k-340) to *Puccinia recondita* and *Blumeria graminis Aegilops* accessions were analyzed. A graphical representation of metabolomics profiles is shown in Figure 1.

Monosaccharides and oligosaccharides accounted for most of the metabolites encountered in the seed metabolome of the resistant accession k-1958, 38.8 and 38.1%, respectively. Meanwhile, in the metabolome of the susceptible accession, oligosaccharides were predominant (55.3%), and the occurrence of monosaccharides was much lower, amounting to 4.6%.

Figure 2 shows the key organic acids detected in the seeds of *Ae. tauschii*. Thus, for phosphoric, glyceric, malic, salicylic, erythronic, ribonic, galacturonic (GalUA), gluconic, caffeic, and gulonic acids, the excess of their content in the seed of *Ae. tauschii* k-1958 was more than 2-fold compared to *Ae. tauschii* k-340.

Also, thirteen sugars were detected in the seeds of the studied accessions. In the seeds of the resistant accession, *Ae. tauschii* k-1958, a very high content of glucose was found, which exceeds by 30 times its content in the k-340 accession (Figure 3). Fructose and galactose contents were also observed to be almost 3 and 12 times higher, respectively, in the grain of the resistant accession, *Ae. tauschii* k-1958, compared to the susceptible accession, *Ae. tauschii* k-340. In turn, the susceptible accession showed high concentrations of sorbose, raffinose, and melibiose, more than two times higher than their concentrations in the k-1958 accession. The contents of sucrose, maltose, and stachyose were 1.3, 1.6, and 2.5 times higher in the resistant accession k-1958 (Appendix A and Figure 3).

The free amino acid composition of seeds was represented by eighteen amino acids, among which three were nonproteinogenic (3-hydroxypipecolic, 5-hydroxypipecolic acids, and pipecolic acid). In terms of the total quantitative amino acid content, 3-hydroxypipecolic acid predominated in the grain of both accessions. Among the defined proteinogenic amino acids, valine and glutamine were predominant in accession k-340, and glutamine was predominant in accession k-1958 (Figure 4).

In the composition of metabolome, twelve fatty acids were determined, among which linoleic acid was the most abundant, followed by palmitic and oleic acids. The total content of these acids was slightly higher in resistant *Aegilops* accessions. Among the detected phytosterols, sitosterol was predominant in the grain of both *Ae. tauschii* accessions (Table 1). The resistant accession had 63% higher sitosterol content than the susceptible accession.

Table 1 summarizes the metabolites involved in plant resistance and important for defense against fungal pathogens.

An analysis of the seed metabolome of the studied *Aegilops* accessions showed that their metabolism differs significantly (Figure 5). These results suggest that the increased synthesis of some metabolites plays an important role in the high resistance of the k-1958 accession to pathogens.

### 2.2. Fungal Microbiome (Mycobiome)

The studied *Aegilops* seed accessions were inhabited by micromycetes at 2.12 × 10^8^ ± 1.9 × 10^7^ CFU/g seeds of the susceptible accession k-340 and 8.51 × 10^7^ ± 2.0 × 10^6^ CFU/g seeds of the resistant accession k-1958.

The mycobiome of the susceptible accession k-340 was represented by the following dominant (more than 10%) genera: *Alternaria* (39.2%), *Blumeria* (38.6%), and *Cladosporium* (12.4%) (Figure 6). In the resistant accession, the genera *Cladosporium* (47.1%), *Alternaria* (19.1%), and *Vishniacozyma* (6.2%) were dominant. Pathogenic micromycetes are shown in Table 2.

Frequently occurring micromycetes (1–10%) of the seed mycobiome of seeds of susceptible accession k-340 were represented by the genera *Vishniacozyma* (3.5%), *Parastagonospora* (1.3%), and *Stemphylium* (1.0%). The genera *Vishniacozyma* (6.2%), *Blumeria* (4.0%), *Sporobolomyces* (2.1%), and *Stemphylium* (1.3%) were detected in the seeds of resistant accession k-1958. Representatives of other genera occured in very low numbers (0.1–1%) and belonged to the genera *Acremonium*, *Pyrenophora*, *Sporobolomyces*, *Selenophoma*, *Gibberella*, *Cystofilobasidium*, *Dioszegia*, *Puccinia*, *Beauveria,* and *Fusarium* (Figure 7).

The genus *Alternaria* includes the pathogenic micromycete *Alternaria infectoria*, and the frequency of occurrence of this genus was 15.5% in the resistant accession of *Aegilops* and 30.7% in the susceptible accession. *Blumeria graminis*, which causes wheat diseases, was present in the mycobiome of only susceptible accession k-340 (7.3%), as well as *Puccinia recondita* (0.1%) and *P. striiformis* (0.03%).

The Simpson and Shannon biodiversity indices were calculated for the fungal microbiomes of *Ae. tauschii* seeds (Figure 8). The Simpson index shows that the microbial communities of seeds k-1958 and k-340 are not significantly different from each other in terms of overall diversity. The Shannon index also showed no reliable differences; however, the average value of the index in k-340 accession is slightly higher than that in k-1958, thus indicating a greater number of dominants in the microbiome and a more uniform distribution of them.

The Venn diagram of the OTUs distribution in fungal seed microbiomes shows that the accession k-1958 has two times more unique OTUs than the k-340 accession (Figure 9).

The analysis of the beta diversity of the fungal communities showed that the accessions of the studied *Ae. tauschii* formed separated clusters (Figure 10).

## 3. Discussion

Seeds of *Ae. tauschii* accessions from the VIR collection differing in their resistance to brown rust were analyzed earlier, and differences in their metabolomic profile were revealed [28]. In particular, it was shown that resistant accessions of *Ae. tauschii* were characterized by the presence of high concentrations of secondary metabolites, including pipecolic acid, stigmasterol, azelaic acid, and pyrogallic acid [28].

In this study, the analysis of the seed metabolome of the studied *Aegilops* accessions showed that their metabolism differed significantly. The metabolomic profiles of the accessions differed significantly between each other in the content of non-protein amino acids, phytosterols, polyatomic alcohols, acylglycerols, acylglycerols, mono- and oligosaccharides, glycosides, phenolic compounds (hydroquinone, kaempferol), etc. (Figure 1; Appendix A). 

The metabolome of the resistant accession k-1958 is characterized by the prevalence of organic acids—phosphoric, malic, ribonic, galacturonic, and gluconic acids—involved in the main metabolic processes of the plant cell (Figure 2a,b). Numerous organic acids also act as antibacterial agents [32]. For example, galacturonic, jasmonic, and salicylic acids act as plant hormones by increasing the synthesis of phenolic compounds to protect plants against pathogens [33]. 

Caffeic acid is a hydroxycinnamic acid that contains both phenolic and acrylic functional groups. The action of caffeic acid and its derivatives against bacteria, fungi, and viruses has been well studied [34,35]. Pyrogallol has a proven antimicrobial action [36,37,38], and the mechanism of this action is enzyme inhibition [39]. Pyrogallol showed its effect against the fungi *Fusarium oxysporum* [40] and *Candida albicans* [41]. In our study, the resistant accession of *Aegilops aegilops* k-1958 was found to have a 3–4 times higher content of pyrogallol and caffeic acid than the susceptible one (Figure 2b). Most of the tested organic acids with antibacterial properties were higher in the resistant accession k-1958 (Figure 2a,b).

Salicylic acid (SA; the main hormone of plant innate immunity) is synthesized from phenylalanine with benzoate as the immediate precursor [42]. The role of SA in plant defense activation is well studied. SAR (systemic acquired resistance) is activated through the combined action of SA and pipecolic acid [43,44]. Hydroxylation of SA leads to the formation of 2,3- and 2,5 dihydroxybenzoic acid (2,3-DHBA and 2,5 DHBA) [45]. Infection with pathogens causes the accumulation of salicylic acid and azelaic acid in the apoplast and symplast, respectively [46,47]. SA in plants has an antifungal activity [48]. Starting from the receptivity of phytopathogen-signaling molecules at the cell membrane, all metabolic processes are controlled by resistance genes that re-regulate a set of defense responses. Glycerol-3-phosphate and N-Hydroxy-Pipecolic Acid, along with SA, also induce SAR in plants [49,50]. 

SA plays a key role in plant immunity, as described above, and yet some pathogens can inhibit SA synthesis in the plant by disrupting SA signaling pathways [51].

We demonstrated that total monosaccharides were significantly (15.4 times) higher in the resistant accession, while total oligosaccharides were almost the same in the two compared accessions (Appendix A). Monosaccharides are a substrate for glycolysis, which increases energy yield by utilizing pyrophosphate instead of adenosine triphosphate (ATP) [52]. Fructose functions as a regulatory metabolite and interacts with plant hormones [53]. Glucose is a versatile carbon source and also acts as a signaling molecule that modulates various metabolic processes in plants [54]. According to our results, the fructose content was 2.8 times higher (Figure 3a), and the glucose content was 30.4 times higher in the seed of the resistant accession (Figure 3b). 

There was no significant difference in oligosaccharides’ accumulation between resistant and susceptible accessions of *Ae tauschii*; however, more significant differences between accessions were found for individual oligosaccharides. Thus, the concentration of sucrose and maltose was higher in the grains of the resistant accession k-1958 (1.3 and 1.6 times, respectively). The contents of melibiose and raffinose were higher in the grains of the susceptible accession k-340 (3.2 and 2.7 times, respectively). The stachyose content was 2.5 times higher in the resistant accession k-1958. Sucrose is included in the stress–plant interaction signaling system and probably reflects the plant’s response to the pathogen [55].

Maltose, melibiose, and stachyose are known to be resistance factors to abiotic stresses such as drought and both high and low temperatures [56,57,58] The higher content of sucrose and stachyose in the grains of the resistant accession Ae tauschii k-1958 is confirmed by our earlier results obtained on a large number of *Ae tauschii* accessions [28]. This probably indicates the participation of maltose and stachyose in the formation of the resistance of *Ae tauschii* to fungal pathogens.

Raffinose belongs to the family of soluble sucrose derivatives, which represent an important form of carbon source in plants. Raffinose metabolism provides readily available energy and carbon to the major metabolic processes in seeds [59]. In our study, the raffinose content was 2.7 times higher in the seeds of resistant accession k-1958 (Figure 3b).

In general, sugars provide energy inputs necessary for plant defense against pathogens, participate in the regulation of “defense” genes as signaling molecules, and are key components of the cellular redox system [60]. An analysis of the role of sugars in providing plant protection has led to the concept of “sweet immunity” and “sugar-enhanced defense” [60,61].

Plant metabolites that act as key components of systemic resistance induction, such as salicylic acid, azelaic acid, pipecolic acid, galactinol, glycerol, and sitosterol [62,63,64,65], were significantly higher in the seeds of the k-1958 *Aegilops* accession (Table 1). The role of galactinol in the systemic resistance of plants against phytopathogenic fungi has been demonstrated [66]. In our study, in the metabolome of the resistant accession k-1958, the content of galactinol was two times higher than in the susceptible accession k-340 (Table 1). This may indicate triggered SAR mechanisms in the resistant accession k-1958 that are regulated by the resistance genes of this genotype through signaling. We did not observe high levels of metabolites involved in SAR in the susceptible accession k-340. 

Different genetic responses of *Ae. tauschii* to the infection with *Puccinia triticina* (causal agent of brown rust) were also obtained for different pathogen-resistant accessions [67]. It was found that, in resistant *Ae. tauschii*, the highest number of DEGs (differentially expressed genes) was associated with the triggering of the immune response, metabolic pathways of jasmonic acid, galactose and hexose, organic and carboxylic acids, and the organization of nucleosomes and chromatin. The up-regulation of “AET6Gv20822700” (6.23-fold) and “AET7Gv21052200” (4.57-fold), encoding the allene oxide cyclase and peptidase families, respectively, led to the activation of JA-dependent signaling cascades, which in turn led to the regulation of the innate immune response [67]. This indicates the activation of induced systemic resistance in the resistant *Ae. tauschii*.

Glycolysis is the first step in breaking down glucose and producing energy for cellular metabolism. The content of major metabolites of glycolysis, such as glucose and pyruvic acid, was higher in the resistant accession k-1958 (Figure 2 and Figure 3). Precursors of important secondary metabolites are formed from major metabolic pathways, such as glycolysis, the tricarboxylic acid cycle, or the shikimate pathway, which is the biosynthetic source of the three aromatic amino acids phenylalanine, tryptophan, and tyrosine [68]. These aromatic amino acids are precursors of plant defense metabolites (Figure 5). Tyrosine and tryptophan are also precursors of a number of new plant defense metabolites (dhurrin and indole glucosinolates, respectively [69]) or can be used by cells as a source of carbon, nitrogen, and energy. In addition, tryptophan is a precursor for the essential phytohormone indole-3-acetic acid. In our study the tyrosine and tryptophan contents were significantly higher in the accession k-1958 (Figure 4 and Figure 5). 

In addition to amino acids, quinones, tocopherols, folates, lignins, and other aromatic compounds are synthesized through the shikimate pathway, which is involved in plant defense [68]. The abundance of metabolites involved in the shikimate and phenylpropanoid pathways was also higher in the resistant *Ae. tauschii* accession k-1958 (Figure 5).

Metabolic pathways of defense against biotic stress include the accumulation of fatty acids. Fatty acids, along with phytosterols, were significantly increased in the metabolic response expressed as resistance to biotic stress [70]. The accumulation of linoleic acid under the action of elicitor (algal polysaccharides) was found to be counteracted by an increase in azelaic acid in tomato plants, as azelaic acid is a regulator of SAR. An increase in palmitic and stearic acids [70], which are involved in the synthesis of cutin and cuticular waxes, which provide protection against pathogens, was also observed [71]. In our study, the k-1958 accession contained higher concentrations of linoleic, oleic and palmitic acids in the seed composition compared with the k-340 accession (Appendix A), which can be responsible or its resistance to the phytopathogenic fungi.

Eight polyols were identified in the seeds of the studied *Aegilops* accessions. Among them, galactinol, glycerol, and myo-inositol were dominant in the accession k-1958 (in descending order of content), and galactinol, arabinitol, and glycerol in the accession k-340. The total value of polyols in the seeds of the resistant accession was 1.6 times higher compared with the sensitive accession, and the inositol isomers (myo- and chiroinositol) were more than three times higher. It was demonstrated that inositol and myo-inositol participate in the plant defense reaction against fungal pathogens [72]. Myo-inositol plays an important role in energy metabolism, as well as membrane and cell-wall synthesis (oligo- and polysaccharides such as raffinose and hemicellulose). Myo-inositol derivatives are secondary messengers for the transmission of various signals in plants [73]. The importance of glycerol [74] and galactinol [66] in plant defense against pathogens has been noted and was proved by our results. The contents of galactinol and glycerol were higher in the seeds of the resistant *Aegilops* accession compared to the susceptible one (by 1.9-fold and 1.3-fold, respectively).

It is known that glycerol synthesis can be activated through the high-osmolarity glycerol (HOG) pathway, which involves two signal transduction chains via the osmosensory histidine kinase proteins SLN1 and SHO1 [75]. Through the HOG pathway and glycerol synthesis, plants can exhibit antifungal defense [74]. In our study, we found that the glycerol content of the resistant accession was 29% higher than that of the k-340 accession (Table 1).

The content of aromatic amino acids (phenylalanine, tyrosine, and tryptophan) was significantly higher in the resistant k-1958 accession (Appendix A), which is consistent with the result about the activation of the organic amino acid pathway in the phytopathogen-resistant k-1958 *Ae. tauschii* accession [67]. Important signals for the regulation of plant responses to environmental changes are transmitted through the metabolic pathways of glutamine and arginine biosynthesis, which is consistent with the findings of Reference [76], which suggested that glutamine plays an important role in plant defense responses through the nitrogen metabolism pathway. 

We found that, among proteinogenic amino acids, valine and glutamine predominated in the k-340 accession, and glutamine predominated in the k-1958 accession (Figure 4), and the accumulation of glutamine (Glu) was almost 3 times higher in the seeds of the resistant accession. Glutamate (Glu) is a precursor of glutamine (Gln) and proline (Pro), and it also serves as a signaling molecule in many physiological processes, including plant defense against cover tissue damage and pathogen invasion [77]. Therefore, we hypothesized that Glu and Pro values would be significantly different in accessions with different degrees of pathogen resistance. However, proline accumulation in the k-1958 and the k-340 seeds was almost the same, and glutamic acid was slightly higher in the k-340. The synthesis of glutamine from ammonia and glutamate is a key reaction of nitrogen metabolism in plants. This reaction is controlled by glutamine synthetase (GS), one of the enzymes involved in mitigating the effects of abiotic stresses on the plant [78]. It is likely that, in our experiment, GS activation and consequent glutamine accumulation occurred under the effect of phytopathogens.

Secondary metabolites are not involved in the main processes of plants. However, these biologically active compounds play an important role in plant defense. They include phenol-containing compounds, carotenoids, and non-proteinogenic amino acids [33,79]. In our study phytosterols were represented by the 4-desmethylsterols group (stigmasterol, campesterol, and sitosterol), and their total content was 1.6 times higher in resistant accession k-1958 (Appendix A). Among the detected phytosterols in the seeds of both *Ae. tauschii* accessions, sitosterol was predominant (Table 1). In the resistant accession k-1958, the content of sitosterol was 63% higher compared to the susceptible accession k-340. Recently, it was demonstrated that sitosterol and stigmasterol play a key role in regulating nutrient flow from the cytoplasm to the apoplast, making plants resistant to pathogens [70]. Phytosterols are involved in membrane biogenesis [80], which affects the defense functions of the plant. The antifungal effect of phytosterols is that the active fat-soluble compounds can easily penetrate the cell wall of fungi, changing its permeability [33]. In addition, phytosterols neutralize reactive oxygen species, preventing their negative effects on cellular structures [70,72].

It is believed that the synthesis of phenolic compounds is more active in resistance plants [72]; however, in our study, the *Aegilops* accessions studied did not differ in this indicator. The dominant phenol-containing compounds of both *Aegilops* accessions were hydroquinone and the kaempferol. The hydroquinone content was higher in the susceptible accession k-340 (87 ppm) compared to the resistant accession k-1958 (78.1 ppm). The kaempferol content did not differ significantly between the two accessions.

The contents of caffeic acid, benzoic acid, its derivatives, and a-tocopherol were higher by two or more times in the resistance accession k-1958 of *Aegilops* compared to the susceptible k-340 (Appendix A). 

It is known that oxy-cinnamic acids can interact with amino groups of aliphatic polyamines to form plant phenylamides, which are a part of the plant defense mechanism against environmental stress factors, including biotic ones [81]. Hence, differences in caffeic acid accumulation in the *Aegilops* accessions with different resistance to the fungal pathogen can be explained by the possible activation of phenylamide synthesis in k-1958 seeds.

The synthesis of phytoalexins is induced a by pathogen attack on the plant, through the activation of β-glucosidase and subsequent release of biocidal aglycones [33]. Presumably, this is caused by the accumulation of hydroquinone and kaempferol-7-O-glucoside and sugar residues in *Aegilops* cereals. The sugar residues (Figure 1) found in higher occurrence in the metabolome of the resistant accession k-1958 may be breakdown products of glycosides. From the group of nucleosides, only adenosine was identified in *Aegilops* cereals, and its absolute content (ppm) was more than 10 times higher in the resistant accession k-1958 compared to the susceptible one, which may be an indicator of the expression of defense-related genes. Comparing the current results with those we obtained earlier [28], we found that the patterns of characterization were mostly repeated. Differences in the total value of polyols and phytosterols of the resistant accession in the composition of the profile of oligosaccharides and polyols were revealed. In both *Ae. tauschii* accessions, sucrose was found to be the dominant oligosaccharide, in contrast to the previously obtained results (when raffinose prevailed in the resistant accessions and sucrose in the susceptible ones).

Examples of the relationship between plant metabolite content and plant microbiome are described in the literature [9,11,82]. The genotype of the resistant *Aegilops* accession accounts for the greater presence of antimicrobial compounds. Thus, the seeds of resistant *Ae. tauschii* accession k-1958 contain more active substances that contribute to high resistance to fungal diseases. 

The seed mycobiome of the pathogen-susceptible accession k-340 was represented by the dominant potentially pathogenic genera *Alternaria*, *Blumeria*, and *Cladosporium*. Both saprotrophic and endophytic, as well as pathogenic, species are found among the fungal genera *Alternaria* [83,84]. The pathogenic micromycetes *A. infectoria* identified in our experiment (frequency of occurrence 15.5% in the resistant accession *Aegilops* k-1958 and 30.7% in the susceptible accession k-340) cause black spot (black point) in wheat [85,86]. *B. graminis* (7.3%), which causes powdery mildew diseases of wheat [87,88,89], was present in the mycobiome of only susceptible accession k-340 (Appendix A). Micromycetes of the genus *Alternaria* have been detected on wheat seeds [86,90].

The genus *Cladosporium* includes both saprotrophic micromycetes and pathogens that cause diseases of cereal crops [85,91,92,93]. We demonstrated that *Cladosporium* sp. dominated in the resistant accession k-1958 (47.1%) compared to the susceptible accession k-340 (11.1%). Potentially pathogenic micromycetes of the genus *Stemphilium* [94] are present in both *Aegilops* seed accessions, at approximately equivalent percentages of occurrence (1.1% in the k-1958 accession and 0.9% in the k-340 accession). In contrast, the potentially pathogenic yeast species *Dioszegia hungarica* [95] was more frequently present in the mycobiome of the resistant accession, but with a very low frequency of occurrence (0.14%).

Other pathogens causing leaf diseases of plants were also detected in the mycobiome of only susceptible accession k-340, including genus *Parastogonospora* (1.15%). The genus *Parastogonospora* was represented in the mycobiome by two species *P. avenae* (0.73%), causing yellow leaf spot predominantly on oats [95,96,97,98] and *P. phragmitis* (0.36%), a pathogen for wild grasses [99]. Pathogenic micromycetes of the genus *Puccinia* [24,100,101] are represented by *P. recondita* (0.1%) and *P. striiformis* (0.03%). The results indicated the presence of pathogenic micromycetes *Blumeria graminis*, *Puccinia recondita* (0.1%) and *Puccinia striiformis* (0.03%) in the sequenced seed mycobiome of only susceptible accession k-340. Potentially pathogenic micromycetes of the genus *Pyrenophora* [102,103,104] were also found only in the seeds of the susceptible accession k-340. 

Our results confirm the visual assessment of *Aegilops tauschii* plant lesions during the 2019 growing season. Leaf rust diseases affected 15.5%, and powdery mildew 75%, of the susceptible accession k-340. At the same time, leaf rust diseases affected less than 5%, and powdery mildew affected less than 10%, of plants of the resistant accession k-1958.

However, representatives of other potentially dangerous genera, namely *Gibberella* [105] and *Fusarium* [106], were detected in the mycobiome of resistant accession k-1958 (0.21 and 0.1%), while their occurrence in the mycobiome of susceptible accession was 0 and 0.01%, respectively. The potential pathogen *Dioszegia hungarica* [95] was found predominantly in the mycobiome of the susceptible accession *Ae. tauschii* k-340 (0.14 and 0.05% respectively). Meanwhile, *Selenophoma linicola*, which is a flax pathogen [107], was present only on an accession of resistant *Aegilops* seeds k-1958 (0.18%). The mycobiome of the resistant accession is characterized by a higher occurrence of saprotrophic microorganisms, many of which are biocontrol agents. Thus, the occurrence of yeasts of the genus *Vishniacozyma* in the mycobiome of the resistant accession k-1958 was 1.8 times higher compared to the susceptible accession k-340. Among them, species of *V. victoriae* and *V. tephrensis* are described as biocontrol agents [108]. Saprotrophic yeasts of the genus *Sporobolomyces* [109] were more frequently found on the resistant accession k-1958. *Sporobolomyces roseus* has also been described as a biocontrol agent [110]. The frequency of the occurrence of *Sporobolomyces roseus* in the resistant accession k-1958 was 5.6 times higher than in the k-340 susceptible accession. However, the occurrence of *Acremonium alternatum*, which is a hyperparasite and biocontrol agent [111,112] was 2.8 times higher on the seeds of the susceptible accession k-340, accounting for 0.67%. Entomopathogen *Beauveria bassiana*, present only on resistant accession k-1958 (0.37%), also exhibited biocontrol activity, preventing tomato and cotton from being infected by *Rhizoctonia solani* and *Pythium myriotylum* pathogens of bulking and root rot [113].

Thus, our hypothesis about the possible presence of antagonist micromycetes in the seeds of brown rust-resistant accession k-1958 and about the lower occurrence of phytopathogens was confirmed. The metabolome of the studied accessions of *Ae. tauschii* with different resistance to brown rust and powdery mildew determines the composition of their mycobiome.

Thus, no rust fungi of the genus *Puccinia* were found on the seeds of *Ae. tauschii* k-1958 accession resistant to brown rust. 

## 4. Materials and Methods

### 4.1. Plant Material

Experimental specimens of *Aegilops* for study were selected while taking into account the analysis of long-term data during their cultivation from 1991 to 2022 at the Dagestan experimental station of VIR and evaluation of their field resistance to the leaf rust pathogen *Puccinia recondita*: k-340—susceptible accession of *Ae. tauschii* ssp. *meyeri* (2n = 14, genome D); and k-1958—highly resistant accession of *Ae.tauschii* ssp. *strangulata* (2n = 14, genome D). In addition, juvenile resistance of the grown accessions was studied when creating artificial infection backgrounds by North Caucasian populations of pathogens of brown, yellow, stem rust, pyrenophorosis, and septoriosis [114]. Biological passports of the studied plant accessions and characteristics of economically useful traits are presented (Appendix A).

The experiment included seeds of *Ae. tauschii* plants obtained during cultivation in field conditions at the experimental field of the Dagestan experimental station of Federal Research Center N. I. Vavilov All-Russian Institute of Plant Genetic Resources (VIR) in 2019, when epiphytotic development of leaf fungal diseases in different phases of plants was observed. This allowed us to completely characterize the accessions of *Ae. tauschii* contrasting in resistance to pathogens.

The experimental station is located in the southern plane zone of Dagestan, in the semi-desert zone of the Primorskaya lowland (N_41.982954, E_48.330189); Khazar village, Derbent district, Dagestan, Russian Federation.

The soils are chestnut, medium humus, and deep columnar Solonets of the heavy loamy variety. The content of humus in the humus horizon is 2–3.5%. The hydrothermal regime of the south-plane zone of Dagestan favors the defeat of barley plants by powdery mildew pathogen due to high air temperatures and humidity [115].

Infestation of plants of the studied accessions was taken into account in the phase of milk-wax ripeness. The degree of damage to plants of k-340 susceptible accession of *Ae. tauschii* in field conditions during the growing season 2019 was brown rust (*Puccinia recondita*) (15%), yellow rust (*Puccinia striiformis*) (0.5%), and powdery mildew (*Blumeria graminis*) (75%). Leaf rust diseases affected less than 5% of plants, and powdery mildew affected less than 10% of plants of the resistant accession k-1958. The evaluation of indicators was determined by scales (Appendix A).

### 4.2. Metabolome Analysis

Metabolomic profiles of *Ae. tauschii* seeds accessions were studied in 5 biological and 3 analytical replications [116]. The seeds were cleaned of glumes and ground. Then, 50 mg of the flour of an accession was homogenized with 500 μL of methanol. After that, 100 μL of the extract was evaporated to dryness with the help of a CentriVap Concentrator (Labconco Corporation, Kansas City, MI, USA). The dry residue was silylated using bis (trimethylsilyl) trifluoroacetamide at 100 °C for 40 min. The trimethylsilyl ethers of the metabolites were separated on an HP-5MS 5% phenyl-95% methyl polysiloxane capillary column (30.0 m, 250.00 μm, 0.25 μm) on an Agilent 6850 gas chromatograph with an Agilent 5975B VL MSD quadrupole mass-selective detector (Agilent Technologies, Santa Clara, CA, USA). The analysis was performed at an inert gas flow rate through the column of 1.5 mL/min. The column was heated from +70 to +320 °C at a heating rate of 4 °C/min. The temperature of the mass spectrometer detector was +250 °C, and the injector temperature was +300 °C. The volume of the injected accession was 1.2 μL. Pyridine solution of triclosan (1 μg/μL) served as an internal standard.

### 4.3. Extraction of DNA, PCR and Sequencing

#### 4.3.1. Isolation of Epiphytic Fungi

Epiphytic micromycetes were isolated according to a previously described modified method [117]. To isolate epiphytic microflora, seed scales were removed from seeds, and 10 g of seeds was placed in a 250 mL Erlenmeyer flask with 100 mL of distilled water. Seeds were shaken at 150 rpm for 1 h. Then, seeds were removed, and liquid fractions were centrifuged at 4000× *g* for 15 min. Pellets were collected and subjected to DNA extraction. DNeasy Plant Pro Kit (Qiagen, Venlo, The Netherlands) was used for DNA extraction from accessions according to the manufacturer’s instructions.

#### 4.3.2. Sequencing ITS

Taxonomic analysis of the fungal community was determined based on the analysis of amplicon libraries of fragments of fungal ribosomal operons (ITS2) obtained by PCR using ITS3/ITS4 primers (GCATCGATGAAGAAGAACGCAGC/TCCTCCGCTTATTATTGATATATATGC). All primers had service sequences containing linkers and barcodes (required for Illumina sequencing). PCR was performed in 15 μL of a reaction mixture containing 0.5–1 unit of Q5^®^ High-Fidelity DNA Polymerase (NEB, Ipswich, MA, USA) activity, 5 pkM each of forward and reverse primers, 10 ng of DNA matrix, and 2nM of each dNTP (LifeTechnologies, Carlsbad, CA, USA). The mixture was denatured at 94 °C for 1 min, followed by 25 cycles: 94 °C for 30 s, 55 °C for 30 s, and 72 °C for 30 s. Final elongation was performed at 72 °C for 3 min. PCR products were purified according to the Illumina recommended method using AMPureXP (BeckmanCoulter, San Diego, CA, USA).

The libraries were further prepared according to the manufacturer’s MiSeq Reagent Kit Preparation Guide (Illumina, San Diego, CA, USA). Libraries were sequenced according to the manufacturer’s instructions on an Illumina MiSeq instrument (Illumina, San Diego, CA, USA), using the MiSeq^®^ Reagent Kit v3 (600 cycle) with double-store reads (2 × 300n). ITS2 library processing was performed using the dada2 package, which performed sequence quality filtering, denoising, ASV (amplicon sequence variant) acquisition, chimera filtering, and taxonomic classification using the UNITE database [118]. The results of taxonomic analysis of the libraries are presented using the QIIME package [119].

#### 4.3.3. Quantitative PCR

Quantitative PCR testing was performed using an PowerTrack SYBR Green Master Mix (Thermo Fisher Scientific, Waltham, MA, USA) on a QuantStudio5 (Thermo Fisher Scientific). The thermocycler parameters were as follows: hold for 2 min at 95 °C, followed by 15 s at 95 °C, 60 s at 60 °C, for 40 cycles. PowerTrack SYBR Green Master Mix reagents (Thermo Fisher Scientific) were used for PCR according to the manufacturer’s recommendations. DNA was used for PCR reactions in 10-fold dilution, the reaction volume was 10 µL, and reactions were performed in 3-fold replicates. The amount of ITS per 1 g of accession was determined using the primer sequence ITS1f TCC GTA GGT GAA CCT GCG G/5.8s CGC TGC GTT CTT CAT CG [120]. The analysis was performed in triplicate. Raw sequences are available in SRA under the accession number PRJNA1145905. 

### 4.4. Systemic Analysis of Fungal Community and Metabolic Pathways

The results of the taxonomic analysis of the libraries are presented using the QIIME package [119]. The fungal communities were assessed with the help of ecological biodiversity indices: the Shannon index and the Simpson index. Principal coordinate analysis (PCoA) was conducted to determine the overall differences in community compositions. Beta diversity analysis was used to assess the richness and diversity of fungal communities.

### 4.5. Statistical Analysis

The results are presented as the means of three replicates with standard error (SE). The data were statistically evaluated using STATISTICA-10 (SPSS, Inc., Chicago, IL, USA) Comparisons with *p* < 0.05 were considered as significantly different. The spread of values is shown as error bars representing standard errors of the means in all the figures. 

## 5. Conclusions

Our studies confirmed the difference in the seed metabolomic profiles of the studied accessions of *Ae. tauschii* which differed in field conditions in regard to their resistance to leaf rust and powdery mildew fungi. The resistant accession of *Ae. tauschii* ssp. *strangulata* k-1958 had a higher content of the main metabolites of glycolysis—glucose and pyruvic acid. The higher content of sucrose and stachyose in the grains of the resistant accession *Ae tauschii* k-1958 probably indicates the participation of maltose and stachyose in the formation of resistance of *Ae tauschii* to fungal pathogens. The content of plant metabolites acting as key components of systemic resistance induction, including salicylic, azelaic, and pipecolic acids, as well as galactinol, glycerol, and sitosterol was also significantly higher in the seeds of the resistant accession k-1958.Differences in the metabolome of *Ae. tauschii* seeds provided a different mycobiome of epiphytic micromycetes. The genera *Alternaria, Blumeria*, and *Cladosporium* dominated on the seeds of the mycobiome of susceptible accession *Ae. tauschii* ssp. *meyeri* k-340. The genera *Cladosporium* and *Alternaria* dominated on the seeds of resistant accession k-1958. Pathogens causing leaf diseases of plants were also found in the mycobiome of only the susceptible accession *Ae. tauschii* ssp. *meyeri* k-340, including *Parastogonospora* (1.15%) and *Puccinia* (0.14%). The mycobiome of the resistant accession k-1958 is characterized by a higher occurrence of saprotrophic microorganisms, many of which can be defined as potential biocontrol agents. These results on the composition of epiphytic micromycetes in the seed mycobiome of *Ae. tauschii* are preliminary and should be confirmed using a large number of seeds from diverse accessions differing in susceptibility to fungal diseases.

## Figures and Tables

**Figure 1 plants-13-02343-f001:**
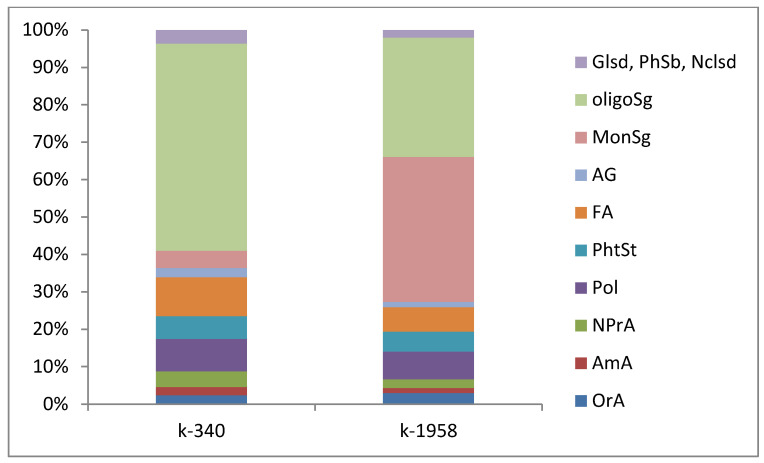
Metabolomic profiles of *Ae. tauschii* seeds differing in resistance to *Puccinia recondita* and *Blumeria graminis.* Notes: k-1958—resistant accession of *Aegilops tauschii* ssp. *strangulata*; k-340—susceptible accession of *Ae. tauschii* ssp. *meyeri.* OrA—organic acids; AmA—amino acids; NPrA—non-proteinogenic amino acids; Pol—polyols; PhtSt—phytosterols; FA—fatty acids; AG—acylglycerols; MonSg—monosugars; oligoSg—oligosugars; glsd—glycosides; PhSb—phenolic substances; Nclsd—adenosine. Data are the sum of three independent experiments and expressed in percentages.

**Figure 2 plants-13-02343-f002:**
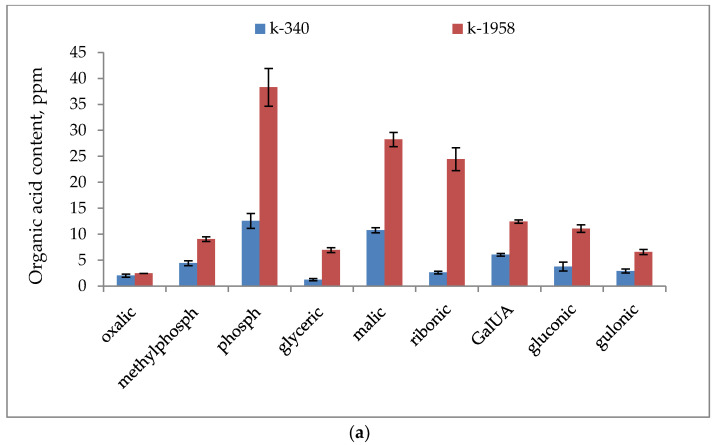
Organic acids content in seeds of *Ae. tauschii* differing in resistance to *Puccinia recondita* and *Blumeria graminis.* Notes: k-1958—resistant accession of *Ae. tauschii* ssp. *strangulata*; k-340—susceptible accession of *Ae. tauschii* ssp. *meyeri.* (**a**) Oxalic, methylphosphonic, phosphonic, glyceric, malic, ribonic, D-galacturonic, gluconic, and gulonic acids; (**b**) pyrogallol, pyruvic, nicotinic, salicylic, and caffeic acids. Data represent the means of three replicates. Bars show ± SE (standard error).

**Figure 3 plants-13-02343-f003:**
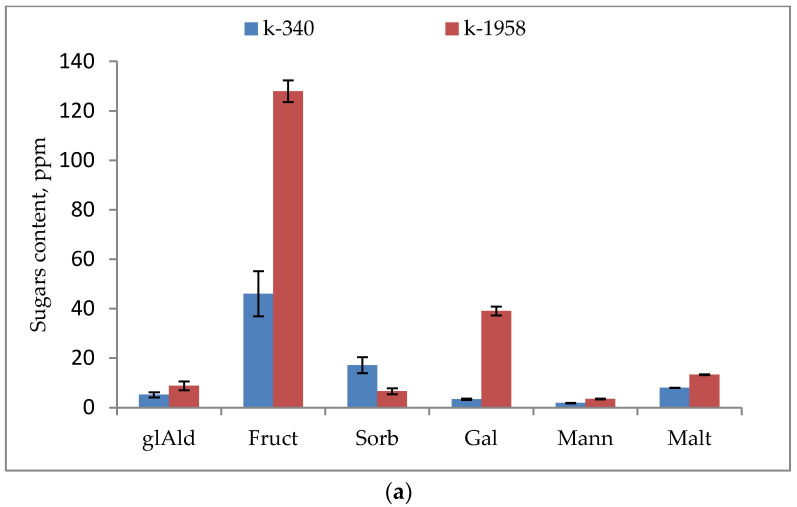
Sugars content in *Ae. tauschii* seeds differing in resistance to *Puccinia recondita* and *Blumeria graminis.* Notes: k-1958—resistant accession of *Ae. tauschii* ssp. *strangulata*; k-340—susceptible accession of *Ae. tauschii* ssp. *meyeri*. (**a**) glAld—glyceraldehyde; Fruct—fructose; Sorb—sorbose; Gal—galactose; Mann—mannose; Malt—maltose. (**b**) Gluc—glucose; Suc—sucrose; Raff—raffinose. Data represent the means of three replicates. Bars show ± SE (standard error).

**Figure 4 plants-13-02343-f004:**
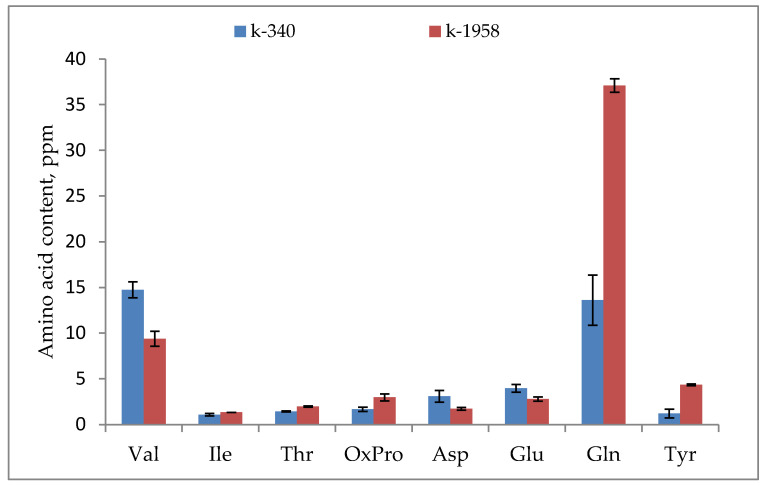
Free amino acids content in *Ae. tauschii* seeds differing in resistance to *Puccinia recondite* and *Blumeria graminis.* Notes: k-1958—resistant accession of *Ae. tauschii* ssp. *strangulata*; k-340—susceptible accession of *Ae. tauschii* ssp. *meyeri*. Val—valine; Ile—isoleucine; Thr—threonine; OxPro—oxyproline; Asp—aspartic acid; Glu—glutamic acid; Gln—glutamine; Tyr—tyrosine. Data represent the means of three replicates. Bars show ± SE (standard error).

**Figure 5 plants-13-02343-f005:**
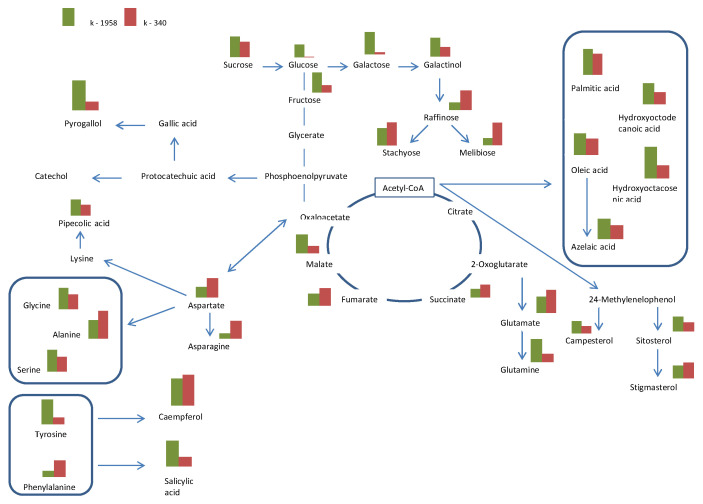
Metabolic pathways in *Aegilops tauschii* seeds differing in resistance to *Puccinia recondita* and *Blumeria graminis.* Notes: k-1958—resistant accession of *Ae. tauschii* ssp. *strangulata*; k-340—susceptible accession *Ae. tauschii* ssp. *meyeri*. The graphic was constructed using the KEGG Database [29].

**Figure 6 plants-13-02343-f006:**
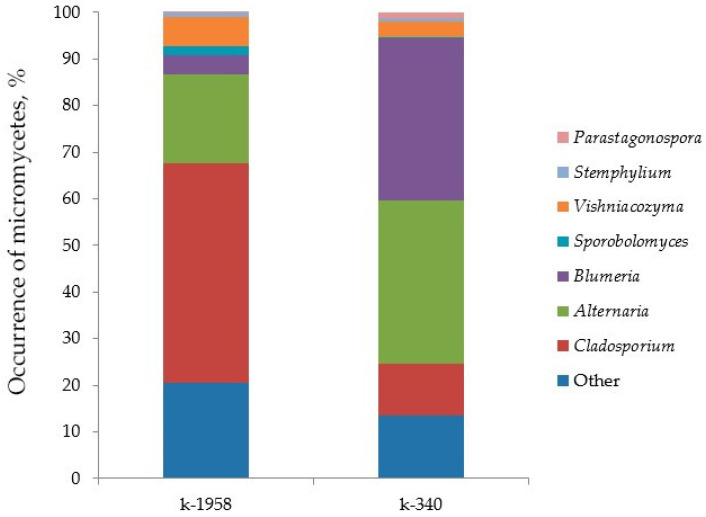
Taxonomic profile of fungal microbiome of *Ae. tauschii* seeds differing in resistance to *Puccinia recondita* and *Blumeria graminis.* Notes: k-1958—resistant accession of *Ae. tauschii* ssp. *strangulata*; k-340—susceptible accession *Ae. tauschii* ssp. *meyeri*. Y-axis shows the occurrence of micromycetes in percentages.

**Figure 7 plants-13-02343-f007:**
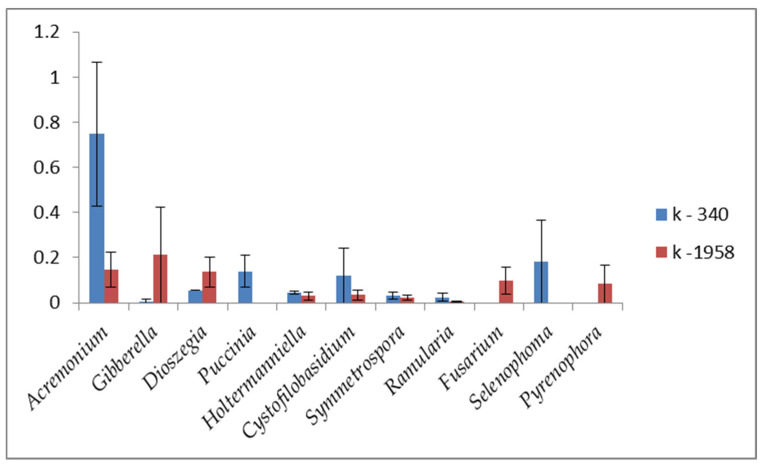
Minor abundance of fungal microbiome of *Ae. tauschii* seeds differing in resistance to *Puccinia recondita* and *Blumeria graminis.* Notes: k-1958—resistant accession of *Ae. tauschii* ssp. *strangulata*; k-340—susceptible accession of *Ae. tauschii* ssp. *meyeri*. Y-axis shows the occurrence of micromycetes in percentages.

**Figure 8 plants-13-02343-f008:**
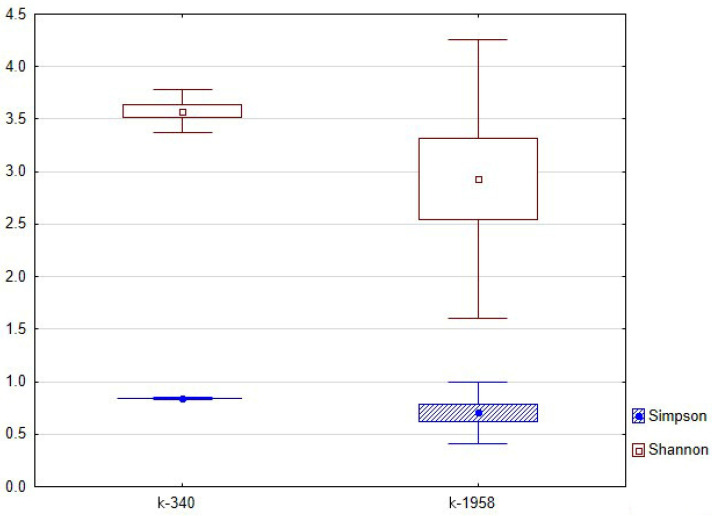
Biodiversity indices of fungal microbiome of *Ae. tauschii* seeds differing in resistance to *Puccinia recondita* and *Blumeria graminis.* Notes: k-1958—resistant accession of *Ae. tauschii* ssp. *strangulata* k-1958; k-340—susceptible accession of *Aegilops tauschii* Coss ssp. *meyeri* k-340. Blue boxes relate to Simpson indices, and red boxes to Shannon indices.

**Figure 9 plants-13-02343-f009:**
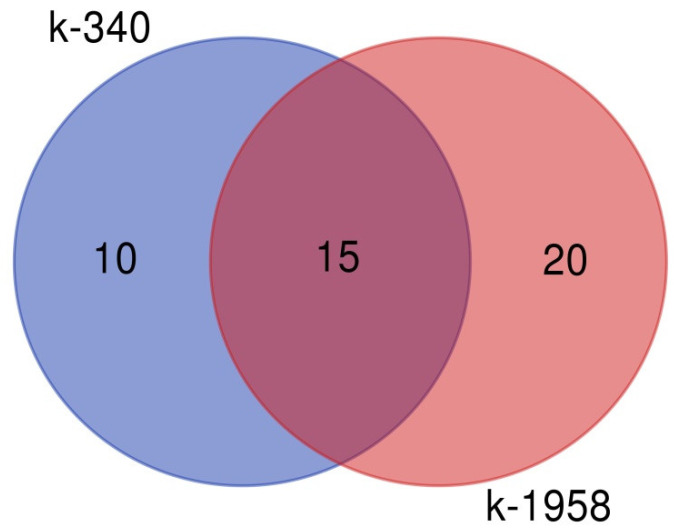
Venn diagram of fungal microbiome of *Ae. tauschii* seeds differing in resistance to *Puccinia recondita* and *Blumeria graminis.* Notes: k-1958—resistant accession of *Ae. tauschii* ssp. *strangulata*; k-340—susceptible accession of *Ae. tauschii* ssp. *meyeri*. The diagram was drawn using the site [31].

**Figure 10 plants-13-02343-f010:**
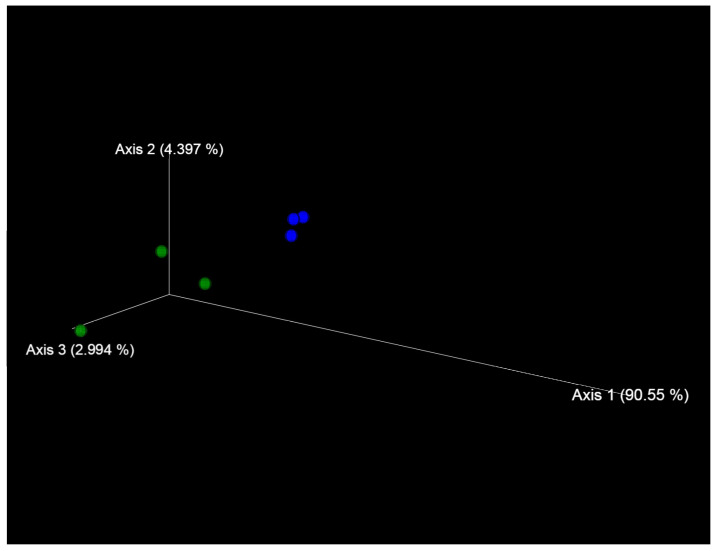
Beta diversity of fungal microbiomes of *Ae. tauschii* seeds differing in resistance to *Puccinia recondita* and *Blumeria graminis.* Notes: k-340—susceptible accession of *Ae. tauschii* ssp. *meyeri*—is marked by blue circles; k-1958—resistant accession of *Ae. tauschii* ssp. *strangulata*—is marked by green circles.

**Table 1 plants-13-02343-t001:** Contents of metabolites involved in plant resistance in *Ae. tauschii* seeds differing in resistance to *Puccinia recondita* and *Blumeria graminis*.

Metabolites	Metabolites Contents, ppm	
	k-340	k-1958
Salicylic acid *	0.13 ± 0.03	0.36 ± 0.04
Pyrogallol *	0.11 ± 0.02	0.40 ± 0.03
Azelaic acid *	1.86 ± 0.20	2.73 ± 0.17
Pipecolic acid *	0.34 ± 0.01	0.51 ± 0.03
Glycerol *	59.68 ± 3.98	76.71 ± 3.21
Galactinol *	77.58 ± 2.80	151.05 ± 8.82
Sitosterol	186.34 ± 26.21	304.23 ± 10.27

Notes: * marked metabolites involved in SAR (systemic acquired resistance); k-1958—*Ae. tauschii* ssp. *strangulata*, resistant accession; k-340—*Ae. tauschii* ssp. *meyeri*, susceptible accession.

**Table 2 plants-13-02343-t002:** The pathogenicity of the highly and medium abundant operational taxonomic units (OTUs) in *Aegilops tauschii* seeds differing in resistance to *Puccinia recondita* and *Blumeria graminis*.

Blast ID	Phylum	Genus Anamorph/Teleomorph	Disease in Wheat	Relative Abundance, %	
				k-1958	k-340
*Cladosporium*	*Ascomycota*	*Cladosporium*/*Davidiella*	Non-pathogenic/black head mold/black point smudge	47.14 ± 22.11	12.35 ± 3.07
*Alternaria infectoria*	*Ascomycota*	*Alternaria*	Black point	15.53 ± 5.19	30.67 ± 2.43
*Blumeria graminis*	*Ascomycota*	*Blumeria*	Powdery mildew	0	7.30 ± 1.99
*Vishniacozyma*	*Basidiomycota*	*Vishniacozyma*	Non-pathogenic	6.15 ± 3.35	3.58 ± 0.49
*Sporobolomyces roseus*	*Basidiomycota*	*Sporobolomyces*	Non-pathogenic	2.06 ± 1.55	0.40 ± 0.04
*Parastogonospora*	*Ascomycota*	*Parastogonospora/Phaeosphaeria*	Spot blotch	0	1.28 ± 0.83

Notes: k-1958—resistant accession of *Ae. tauschii* ssp. *strangulata*; k-340—susceptible accession of *Ae. tauschii* ssp. *meyeri*. Wheat diseases were selected using the site the American Phytopathological society [30].

## Data Availability

Data are contained within the article and Appendix A.

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
