# Peer review of "Metabolome and Mycobiome of Aegilops tauschii Subspecies Differing in Susceptibility to Brown Rust and Powdery Mildew Are Diverse"

_plants, 2024, doi:10.3390/plants13172343_

Round 1

Reviewer 1 Report

Comments and Suggestions for Authors

 In the manuscript named “Metabolome and mycobiome assessment of Aegilops tauschii Coss seeds varied in resistance to the plant leaf diseases”, Veronika N. Pishchik et al have performed integration analysis including metabolome analysis, and metagenomes analysis, their results have shown higher content of glucose and organic acids with positive roles in determining tolerance of brown rust and powdery mildew. These findings were valuable and helpful for wheat genetic improvement in future. The manuscript was well prepared, but there were some comments about it.

(1) The conclusion was too redundance, please remove little-issues, or not-key words, and integrate the conclusion in single or two paragraphs.

(2) Please add longitude and latitude of experiment field.

(3) The Illumina MiSeq data or reads should be submitted to public database with accession number in manuscript.

(4) All metabolic analysis were detected with chemical compound, could authors investigate some key genes in these metabolic pathways, see figure 5, or supply more molecular evidence for their conclusion, such as qRT-PCR, etc.

(5) The figures should be kept in step with each other, figure 1 has different fonts with figure 2, and the y-axis has missed legend, other figures had similar comments, please check them thoroughly.

(6) The sample names should be identical in manuscript, including all text, and figures, authors have used k-340 or 340 in different text or figures.

(7) Authors have discovered there were more oligosugars in k-340 than k-1958, but they didn’t investigate this finding in detail, while the monosugars have been well analyzed, why did they miss oligosugars results?

Author Response

Dear reviewer 1,

Thank you for attention to our article and its detailed analysis.

All your comments have been taken into account and corrections have been made.

(1). The conclusion was too redundance, please remove little-issues, or not-key words, and integrate the conclusion in single or two paragraphs.

1.A. The text of the conclusions has been abbreviated Please, see lines 652-672

(2). Please add longitude and latitude of experiment field.

2.A. We added longitude and latitude of experiment field  Please, see line 571

(3) The Illumina MiSeq data or reads should be submitted to public database with accession number in manuscript.

  1. The Illumina MiSeq data were submitted to NCBI database Please, see line 638

(4) All metabolic analysis were detected with chemical compound, could authors investigate some key genes in these metabolic pathways, see figure 5, or supply more molecular evidence for their conclusion, such as qRT-PCR, etc.

4.A.Thank you for attention to our article and its detailed analysis. The current study was limited by the funding of the foundation mentioned in the manuscript, so now we present a part of the work, which is planned to be continued. The conclusions are preliminary and after the completion of this project we have just planned to expand the research with the inclusion of molecular methods and investigate some key genes in these metabolic pathways.

 (5) The figures should be kept in step with each other, figure 1 has different fonts with figure 2, and the y-axis has missed legend, other figures had similar comments, please check them thoroughly.

5A. We have corrected the figures and captions according to the reviewer's comments. We added the Y-axis legeng. Please, see lines 216, 240

(6) The sample names should be identical in manuscript, including all text, and figures, authors have used k-340 or 340 in different text or figures.

6A. We have corrected the figures and captions according to the reviewer's comments

(7) Authors have discovered there were more oligosugars in k-340 than k-1958, but they didn’t investigate this finding in detail, while the monosugars have been well analyzed, why did they miss oligosugars results?

7 A. In the corrected article, we analyzed oligosaccharides (Please, see lines 139-148 p.4-5 and 329-343 and 349-353)

Reviewer 2 Report

Comments and Suggestions for Authors

The Authors compared the methabolomic profiles and the mycobiomes of seeds of two subspecies of  Aegilops tauschii differing in susceptibility to foliar fungus pathogens to verify the hypothesis that the metabolomic profile and mycobiome characteristics of the two accessions are correlated to the resistance/susceptibility  The very restricted number (just two) of examined accessions to not allow to generalize. the cocnclusions. However the Authors are conscious of the limit of their study and commented it. The paper is clearly presented and consequential. The Discussion is very detailed and appropriate. Most of my comments and corrections concern formal aspects.

More in detail:

- Use the terms accession or  ssp. instead of sample  troughout the text  as they are more appropriate

- Cite the name of the Author of latin names of the species just once when the latin name is mentioned for the first time in the text.

- Cite the extended latin name of species only when it is first mentioned for the first time in the text. The report the abbreviated name of the genus .

-Use the term susceptible instead of sensitive referring to fungal infections 

-Merge the 'Notes? into the captions of figures  in

Fig 1 (Lines96-101)

Fig. 2 (Lines 117-121)

Fig. 3 (Lines 125-139)

Fig. 4 (Lines 149-153)

Fig. 5 (Lines 173-175)

Fig. 6 (Lines 186-187)

Fig- 7 (Lines 204-205)

Fig. 8 (Lines 221-223)

Fig. 9 (Lines 229-231)

Fig 10 (instead of Fig. 9) (Lines 237-239) and correct corresponding citation in the text.

- For more detailed corrections and comments see notes in the text (attached file)

Comments on the Quality of English Language

The English style requires minor refinements

Author Response

Dear reviewer 2,

Thank you for attention to our manuscript and its detailed analysis.

All your comments have been taken into account and corrections have been made.

Our findings are preliminary and we plan to expand the scope of the experiment in the future with a wider representation of Ae.tauschii  accessions and molecular research methods.

- Use the terms accession or  ssp. instead of sample  troughout the text  as they are more appropriate

We use the terms accession in the corrected manuscript.

- Cite the name of the Author of latin names of the species just once when the latin name is mentioned for the first time in the text.

Done

- Cite the extended latin name of species only when it is first mentioned for the first time in the text. The report the abbreviated name of the genus .

Done

-Use the term susceptible instead of sensitive referring to fungal infections 

Done.

-Merge the 'Notes? into the captions of figures  in……

Done 

Round 2

Reviewer 1 Report

Comments and Suggestions for Authors

Thanks for authors’ work, most of my comments were systematically addressed, but there were also some minor issues, such as figure 3, the “Fig 3A” would be recorrected as “A”, the table 1 had missed one line, figure 6 had missed y-label…Please check them. Good luck.

Author Response

Dear reviewer 1,

All your comments have been taken into account and corrections have been made.

1.“Fig 3A” replaced by “A”. Similar changes are made in Fig. 2

2.The bottom line in table 1 is restored

  1. In Figure 6, the signature of the y-axis has been added.

All changes are labeled in yellow.

I apologize for the technical inaccuracies.

Sincerely

Dr. Pishchik V.N.

Round 3

Reviewer 1 Report

Comments and Suggestions for Authors

Thanks for authors’ work, the revision was OK, I have no new comments about it. Good luck.